# Exploring stakeholders' experiences and perceptions regarding barriers to effective surveillance of communicable diseases in a rural district of Pakistan: a qualitative study

Imran Naeem ,[1] Sameen Siddiqi ,[1] Amna Rehana Siddiqui,[1] Rumina Hasan[2]

¹Department of Community Health Sciences, Aga Khan University, Karachi, Pakistan
²Department of Pathology & Laboratory Medicine, Aga Khan University Hospital, Karachi, Pakistan

**Correspondence to**
Dr Imran Naeem;
imran.naeem@aku.edu

## ABSTRACT

**Objective** To explore the experiences and perceptions of health system stakeholders of a rural district of Sindh, Pakistan regarding the barriers to effective surveillance of communicable diseases.

**Design** This qualitative descriptive exploratory design comprised in-depth interviews. Both inductive and deductive thematic analysis was applied to identify key themes from the data.

**Settings** The study was conducted in public sector healthcare facilities and the district health office of the rural district of Thatta, in Sindh province, Pakistan.

**Participants** Fifteen healthcare managers and healthcare providers working in the eight public sector primary and secondary healthcare facilities were interviewed using an open-ended in-depth interview guide.

**Results** Key themes that emerged from the data were: poor governance and absence of surveillance policy framework; fragmentation in the health system leading to lack of uniform reporting; inadequate (human) resources that weakened the infrastructure for disease surveillance; hospital-based reporting of cases that led to a predominantly passive surveillance system; paper-based surveillance system as the key determinant of delayed reporting; non-utilisation of surveillance data for decision making; absence of local laboratory capacity to complement the detection of disease outbreaks and lack of private sector integration in disease surveillance.

**Conclusions** Poor governance and lack of policy framework were perceived to be responsible for weak surveillance infrastructure. Resource deficiencies including inadequate human resource, paper-based reporting and the absence of local laboratory capacity were considered to result in delayed, poor quality and incomplete reporting. The lack of private sector engagement was identified as a major gap.

## BACKGROUND

Infectious diseases continue to pose threat to the health of the public globally. In low-income and middle-income countries, infectious diseases form a significant portion of the disease burden including HIV/AIDS, tuberculosis (TB), malaria, respiratory infections, hepatitis B & C in adults and pneumonia and diarrhoea in children under 5 years of age.[1 2] In Pakistan, communicable diseases remain a major cause of public health concern, with a significant contribution to morbidity and mortality. Conditions such as overcrowding, low socioeconomic status, poor hygiene and unsafe drinking water and poor awareness of health lead to an environment conducive to disease outbreaks. The Health System of the country is overwhelmed by issues of poor governance and lack of resources, resulting in a surveillance system that is ill equipped at detecting outbreaks of infectious disease.[3] Before the devolution of the health system, Pakistan had two main sources to collect data for health indicators, namely: (1) health management information system: this was designed to collect data on selected health indicators from health facilities with established reporting

## STRENGTHS AND LIMITATIONS OF THIS STUDY

⇒ The study has explored experiences and perceptions regarding barriers to effective surveillance of communicable diseases in-depth by involving representatives from various levels of the healthcare system, including from the public health sector and those working under public–private partnership.

⇒ The inclusion of both healthcare managers and healthcare providers in the study provided deeper insights into barriers at both the stewardship level and the operational level.

⇒ The study is among few in Pakistan to adopt a qualitative research approach for exploring barriers to infectious disease surveillance as perceived and experienced by health system stakeholders.

⇒ The study was conducted in a rural district of a developing country, hence study findings should be interpreted in a similar context.

lines from provincial to federal health ministry; (2) data from vertical programmes such as national TB control programme, malaria control programme, HIV/AIDS control programme among others that also reported to federal health ministry.[4] Following the health system devolution in 2010, administrative powers were devolved to provinces with the district becoming the autonomous unit for defining its health priorities and health planning.[5] In the new, although ill-prepared administrative setup district health information system (DHIS) was established to collect disease-related data from the health facility level.[6] In all provinces of the country including the Sindh province, DHIS is the only source of information on health indicators of the population based on service delivery data from public sector health facilities. However, the DHIS has remained underused for communicable disease surveillance. In the last 5 years, rural districts of Sindh have witnessed outbreaks of diseases such as measles[7] and HIV/AIDS[8] while the surveillance system was unable to predict these outbreaks. As per the international health regulations, member states of the WHO have the obligation to develop and strengthen disease surveillance with the help of existing health system resources.[9] However, the emergence of recent outbreaks has indicated weaker surveillance of communicable diseases, more so in rural areas of Pakistan. Moreover, only diseases with global priority (such as COVID-19 and Polio) have managed to get attention, whereas surveillance for diseases of national priority has often struggled to compete for policy space and resources. The scientific literature on infectious disease surveillance in Pakistan is either quantitative in nature or mostly has discussed the implementation of models like the disease early warning system for surveillance.[10–12] However, there is a dearth of literature regarding the challenges that healthcare managers and providers face when implementing disease surveillance programmes and perceptions and experiences of these healthcare professionals regarding barriers to effective surveillance of infectious diseases are less well studied in Pakistan. Such information will not only give an in-depth insight into the challenges faced in infectious disease surveillance but also inform the policy makers with recommendations for addressing those challenges. Hence, we conducted this study to explore the experiences and perceptions of health system stakeholders of a rural district of Sindh, Pakistan regarding the barriers to effective surveillance of communicable diseases.

## METHODS

### Study design and setting

This study used a qualitative descriptive exploratory design to explore the perceptions and experiences of district health system stakeholders regarding the barriers to an effective surveillance system for communicable diseases in the rural district of Thatta located in the province of Sindh, Pakistan. Thatta is situated approximately 100 km from the provincial capital of Karachi. It is a predominantly rural district with an approximate population of 1 million.[13 14] Health indicators including maternal mortality ratio and neonatal mortality rate are among the worst in the country.[15 16] The situation of the healthcare system of Thatta is comparable to any other rural district of Pakistan with inadequate infrastructure and resources. There exist primary and secondary healthcare facilities in the district and there is also a private healthcare system in the district comprising general practitioner clinics and small hospitals. The study was conducted from 15 February 2022 to 30 April 2022, in eight public sector primary and secondary healthcare facilities in the district.

### Study participants

We used purposive sampling to select study participants. Eligible participants were healthcare managers and healthcare providers working in the eight public sector primary and secondary healthcare facilities. Healthcare managers were those working at the district level and were responsible for the management of health services. Healthcare providers were the doctors responsible for the provision of clinical care at selected healthcare facilities. Participants included both males (n=12) and females (n=3). The age range of study participants was 29–57 years. The experience of study participants with the district health system ranged from 5 to 30 years.

We used the 'saturation principle' to determine the sample size for the study. On researchers' observation, with further interviews yielding no new or significant findings, data collection was stopped. This resulted in 15 in-depth interviews with healthcare managers and healthcare providers.

### Data collection

The data were collected using an open-ended interview guide. The guide comprised questions and probes regarding perceptions and experiences about the current state of communicable disease surveillance and challenges in infrastructure and financing for surveillance, barriers in data reporting, timeliness of reporting and quality of reported data. The guide was developed following a thorough literature review and the researchers' own experience and expertise on the subject. The guide was piloted before data collection. Based on the pretest results, researchers gained new insights and revised the interview guide by adding further questions and probes. The interviews were conducted by the first author (IN) having research experience in health systems, communicable diseases and qualitative research. The first author moderated the interviews along with note-takers. The interviews were conducted in the local language (Sindhi) and were audiotaped. A debriefing session was held after each interview to reflect on the participants' responses. The interviews took place in health

facilities at a time suitable for study participants, with each interview lasting about 40–50 min.

## Researchers' reflexivity

In qualitative research, research findings are liable to be influenced by researchers' interest and understanding of the topic. In this study, researchers were not the staff of the health facilities where the study was conducted. However, due to the research team's 4 years of work experience in the district, they were not considered outsiders by the study respondents but rather someone familiar with the district health system and interested in exploring their views. Moreover, researchers used reflexive notes during data collection that fed into the interpretation of study findings to minimise researcher bias.

Researchers had knowledge of the health system and infectious diseases research which influenced the development of the interview guide, however, while pretesting the guide researchers gained new knowledge which informed the revision of the interview guide.

## Patient and public involvement statement

Patients or the public were not involved in the design, or conduct, or reporting, or dissemination plans of our research

## DATA ANALYSIS

We used both inductive and deductive methods to perform data analysis. The analysis approach used a combination of interviews with study participants facilitated by field observations. Researchers' background knowledge of health systems and infectious disease research guided the process of data collection and analysis using the deductive method. However, during the analysis, new themes emerged that were analysed using the inductive method. All audio tapes were transcribed from Sindhi to English and interview notes were written. Manual content analysis was performed where interview notes and notes from audio transcripts were read and reread to identify patterns in data. Manual codes were assigned to identified patterns in data which were subsequently grouped and classified into main themes. All the authors and field team members read the themes, and discrepancies were discussed during the interpretation and analysis of data. The themes emerging from the analysis were shared and discussed with study participants for their comments.

## RESULTS

Data analysis identified a number of core themes from in-depth interviews. Where appropriate, direct quotes from the interviews have been used to ensure rigour in the data reporting. To avoid identifying specific individuals, the participants have been allocated aliases. Table 1 summarises the themes and codes that emerged from the data.

## The absence of a surveillance policy framework and poor governance leads to an ill-defined disease surveillance system

All study participants unanimously pointed out the lack of a comprehensive policy framework for disease surveillance. A respondent mentioned that despite that following devolution in the health system where the province has the autonomy of decision making, no initiative regarding disease surveillance policy has been taken. As a result, there is a lack of clarity in reporting disease-related data, coordination between different stakeholders and meaningful analysis of data and its use for taking action.

> National health policy emphasizes the importance of disease surveillance and having such a system in place; however, it is up to provinces to develop detailed guidelines for disease surveillance and ensure its implementation which is not happening! [Participant 14].

### Provincial disease surveillance and response unit

One of the study respondents mentioned that a digital information system named provincial disease surveillance and response unit (PDSRU) was developed in 2016 with the support of donor money. The PDSRU had all the communicable diseases of importance listed and the system was expected to be linked up with secondary hospitals in the province for regular data collection. However, the PDSRU unit has remained dysfunctional and has not been utilised for the purpose it was built for.

> It is very unfortunate that we got the donor money to establish a surveillance system in the province, but we didn't plan for resources to make the system functional [participant 5].

### District health information system

One of the respondents said that the DHIS was developed and implemented in 2010.[6] It was meant to collate service delivery data from primary and secondary health facilities with a monthly reporting frequency. The data are collected on paper at the health facility level and a hard copy of the monthly DHIS reporting form is handed over physically by each facility to the district health office. There, a computer operator enters the data into the digital portal of DHIS which then can be viewed on a dashboard.

In its current state, DHIS is the only functional health information system that has up-to-date disease information. But a study respondent pointed out that this information remains underutilised for disease surveillance as there is hardly any review and feedback on the reported data from higher levels (ie, district health office or provincial health department). Moreover, since the data are

**Table 1** Themes and codes emerging from data

| Themes | Codes |
|---|---|
| The absence of a surveillance policy framework and poor governance leads to an ill-defined disease surveillance system | ▶ Lack of provincial policy on infectious disease surveillance<br>▶ Lack of laws<br>▶ Lack of surveillance standards<br>▶ Lack of resource planning leading to dysfunctional digital information systems<br>▶ Lack of planning to ensure integrated surveillance of infectious diseases |
| Fragmentation in the healthcare system is a hindrance to a uniform reporting system | ▶ Poor coordination between health system stakeholders<br>▶ Lack of integration between different levels of health facilities<br>▶ Lack of defined reporting lines for surveillance data<br>▶ Different organisations managing various levels of health facilities in the district |
| Inadequate resources translate to poor disease surveillance | ▶ Inadequate provision of facilities and equipment<br>▶ Lack of dedicated human resource for surveillance<br>▶ Lack of financial support<br>▶ Lack of dedicated line item for surveillance in provincial/district budget |
| In the current system surveillance is predominantly passive | ▶ Hospital-based surveillance<br>▶ Surveillance data collected from patients presenting to hospitals<br>▶ Lack of human resource to conduct surveillance in communities<br>▶ Low level of surveillance activities outside hospitals |
| Paper-based reporting is a key determinant of delayed disease reporting | ▶ Surveillance data compiled on hard copies<br>▶ Hard copies are delivered from health facilities to the district health office<br>▶ No dedicated human resource to transfer health facility reports to the district health office |
| Surveillance data are underutilised for evidence-based decision making | ▶ Monthly report submission by health facilities to the district health office is mandatory<br>▶ No feedback was provided from the district health office to facilities on submitted reports<br>▶ Submitted reports are not reviewed for data errors<br>▶ No one from the district or provincial makes monitoring visits to check the fidelity of reported data |
| Lack of laboratory testing capacity takes a toll on disease surveillance | ▶ There is no laboratory capable of conducting tests for diseases under surveillance<br>▶ Samples are sent to a regional laboratory for testing |
| The lack of integration of the private sector in disease surveillance is a major gap | ▶ At the provincial or district level, no measures are taken to bring surveillance data from the private health sector into the mainstream<br>▶ The private health sector does not report surveillance data to the district health office or provincial health department except in case of COVID-19 |

collated and shared at the end of each month, its utility for detecting disease outbreaks is limited.

> DHIS data can be a very good source of passive surveillance, but we need to increase the reporting frequency from monthly to weekly and have regular reviews of the reported data [Participant 5].

### Communicable disease control unit

A study respondent mentioned that for infectious diseases, a communicable disease control (CDC) unit was established at the provincial level in 2015. The intent was to integrate all the vertical (disease-specific) programmes under one roof for planning, resource allocation and public health interventions. Under the CDC unit, bloodborne diseases like HIV/AIDS, hepatitis C and hepatitis B could be looked after single-handedly when addressing the mode of spread, designing interventions and surveillance. Unfortunately, at the district level, the CDC unit lacks infrastructure (eg, dedicated building and office space, computers and internet), dedicated human resources (district focal CDC person) and a plan ensuring how the integration of various diseases can take place.

Having CDC presents an immediate opportunity for syndromic surveillance; we should not waste this opportunity [Participant 10].

### Fragmentation in the healthcare system is a hindrance to a uniform reporting system

Under the public–private partnership (PPP), the public sector healthcare system in Thatta has been contracted out to various private providers. Under PPP, these providers are responsible for providing health services on a day-to-day basis while the health department provides a budget.[17]

One of the study respondents mentioned that in the Thatta district extensive contracting out of health services has been done and that each level of health facility (ie, primary and secondary) is managed by a different private partner. Distinct reporting lines and a lack of data-sharing mechanisms between private partners have given rise to fragmentation within the district health system.

> Primary level facilities are supposed to be linked with secondary level facilities for referrals and reporting of disease-related data, however, there are so many partners with each having its reporting line. This has

negatively affected the reporting of disease surveillance [participant 10].

A healthcare manager pointed out that in the Thatta district, surveillance for vaccine-preventable diseases (VPDs) is conducted by the district health office with the district surveillance coordinator as the focal point. Whereas notifiable diseases are reported by the focal persons of the disease-specific programmes with a flow of information that is separate from that of VPDs. This vertical nature of surveillance of different diseases is inefficient in terms of resource utilisation as it creates parallel systems where surveillance activities are carried out in siloes. These inefficiencies in turn create a weaker surveillance system and put the population at higher risk for communicable disease outbreaks.

> Every disease-specific program such as that of tuberculosis, malaria etc reports its data in isolation, how can we have an integrated disease surveillance system in this situation? [Participant 7].

### Inadequate resources translate to poor disease surveillance

A healthcare manager highlighted the dearth of human resource for surveillance in the district. He mentioned that in the district health office, there is only one person (district surveillance coordinator) who is tasked to coordinate for surveillance of VPDs, data reporting and data entry into DHIS for all eight public sector health facilities of the district. The increased workload results in delayed data entry and delay in data transfer to district health stakeholders. Not only that he lacks a team of dedicated individuals for surveillance, but he also has no provision of transportation for reaching out to health facilities and communities for active surveillance. Most of the time, he has to rely on making telephone contact with each in charge of health facilities to get disease-related information.

All the study respondents mentioned that reporting for VPDs is mandatory and health facilities are required to send weekly reports (even for no cases called 'zero reports') of VPDs to the district surveillance coordinator. But due to a lack of resources for example, transportation and dedicated surveillance staff, the reports from various health facilities in the district often get delayed.

> Expecting one person to lead disease surveillance in the district, in the absence of transportation and adequate human resource is too much to ask [Participant 2].

A district healthcare manager pointed out that for notifiable diseases, the district focal persons of disease-specific programmes have the responsibility of reporting data. The primary role of the focal person is programme implementation, which involves numerous tasks from planning and implementing to monitoring and reporting programme activities. In absence of adequate human resource, focal persons primarily rely on data reported as part of programme implementation. This has limitations, namely (1) this is a form of passive surveillance rather than active; (2) reporting of programme data takes place at specific intervals whereas surveillance requires continuous monitoring of disease cases and prompt action. Many study respondents believed that these limitations lead to a disease surveillance system that struggles with the timely detection, and reporting of disease outbreaks.

### In the current system, surveillance is predominantly passive

Study respondents pointed out that since the data collection is passive as it is collected largely from those patients that present to health facilities, hence it is a form of passive surveillance rather than active surveillance.

> We could be missing out on outbreaking at the community level since there is no one going and actively screening the community members [Participant 1]

Respondents attributed the lack of community-based active surveillance to the lack of resources and mentioned that there is a need to increase resource allocation for strengthening active surveillance.

### Paper-based reporting is a key determinant of delayed disease reporting

Several study respondents attributed the delayed reporting of surveillance data to paper-based reporting. The system for reporting health-related data from health facilities to the district health offices in Pakistan is paper-based for both VPDs and notifiable diseases. The health facility in charge fills out a zero-reporting form (for VPDs) or a case report (for a notifiable disease). A physical copy of this report is then sent to the district surveillance officer. This often gets delayed as someone must visit the health facility or district surveillance coordinator to deliver the physical copy of the report. To prevent delay, telephonic contact often plays a key role, however, it is not a feasible way to disseminate the information to all the relevant stakeholders.

> We should have a digital system with a dashboard which can show data in real-time. We will avoid delayed reporting forever [participant 2].

### Surveillance data is underused for evidence-based decision making

Most study respondents mentioned that data verification is essential to ensure its fidelity. To ensure data quality, monitoring and supervision of field staff and the facility healthcare staff sending data need strengthening. But all respondents agreed that in the existing system of disease surveillance, there is an emphasis on data collection only. No monitoring and supervision are happening from the district or provincial levels. Other than the circumstances where a suspected or confirmed case of disease under surveillance is reported, there is no action taken to verify the authenticity and quality of routinely reported data.

Despite that, the data is regularly collected through DHIS and other channels, it is rarely reviewed or utilized for analysis or action [Participant 12].

## Lack of laboratory testing capacity takes a toll on disease surveillance

All the study respondents showed their concern regarding poor laboratory capacity for surveillance in the district. Among the eight public sector health facilities in the Thatta district, only two have clinical laboratories. However, none of these laboratories is equipped to conduct testing for any of the VPDs or notifiable diseases in the district. Biological samples collected from suspected patients are sent to a regional laboratory that is based in the capital city of the country (laboratory at National Institute of Health, Islamabad) situated at least 1000 kilometres from Thatta district.

A healthcare manager pointed out that the absence of a fully equipped laboratory nearby calls for measures to ensure proper storage and transportation of biological samples. This not only is resource-intensive but adds to the delays in the system staggered by issues of delayed reporting.

> Provincial health department should, at the least, take measures to build a laboratory in the province so that disease surveillance can be made a little more efficient [Participant 6].

## The lack of integration of the private sector in disease surveillance is a major gap

All the study respondents considered the absence of private-sector integration as an important gap. The private health sector is a major stakeholder in service delivery and caters to approximately 70% healthcare needs of the population. Unfortunately, however, the service delivery data of the private health sector is not integrated into the DHIS of the public health sector. In fact, the government is still lagging in taking measures to regulate the private health sector in Sindh province.

One of the respondents said that integration of data from the private sector in the health information system is essential to ensure effective disease surveillance as currently, a significant chunk of the population's disease burden remains to be captured.

> Our disease surveillance will be at a loss from capturing the true disease burden unless it integrates data from the private health sector [Participant 6].

## DISCUSSION

Our study explored multifaceted barriers to effective surveillance of communicable diseases in a rural district of Pakistan. Our study showed that the lack of policy guidelines at the provincial level was fundamental to ineffective disease surveillance and poor data reporting. Poor stakeholders' coordination led to a lack of sharing of surveillance data,

hampering the surveillance efforts. Due to poor resource planning, the digital information systems that is, PDSRU and DHIS built using donor resources were underused. Having the district surveillance coordinator as the sole person responsible for surveillance activities in the district, in absence of additional resources, was considered a major resource gap by study respondents. Most study respondents were concerned about the predominantly passive nature of existing district surveillance. Paper-based reporting together with inadequate human resource was considered an important cause of delayed reporting in surveillance. The lack of laboratory testing capacity in the district was another determinant for delayed reporting. There was an increasing emphasis on collecting data than using it for predicting outbreaks or taking measures to control these. The absence of inclusion of infectious disease surveillance data of private sector data in the district surveillance was identified as another major gap by study respondents.

Despite that a decade has elapsed since the health system in Pakistan was devolved, the pace of provinces taking charge of health planning and resource generation has remained rather sluggish. A study respondent mentioned that despite the increasing infectious disease outbreaks in Sindh and other provinces in recent years that is, measles, HIV/AIDS and the global COVID-19 pandemic, initiatives regarding comprehensive guidelines and laws for disease surveillance are still lacking. Studies from Nigeria and Zambia show that the laws related to public health surveillance existed but were considered to be outdated and/or poorly implemented.[18] Studies from Iran and Palestine concluded that having laws and policies for disease surveillance enable governments to allocate funds for establishing surveillance programmes and that health authorities should play a lead role in ensuring their implementation.[10 19]

Having a dysfunctional PDSRU for reporting surveillance data is a classic example of relying on donor money instead of concentrating on building local capacity for a functional health information system. Many respondents termed this unfortunate and emphasised investing resources in reviving the PDSRU. In a study from China inadequacy of resources at the local level was identified as an important determinant of a functional digital information system for disease surveillance. The study reported that the top tier of government invests more in building digital information systems, but lower levels do not receive enough planned resources to ensure its implementation.[20] A functional digital information system has been found to speed up the reporting, improve data flow and ensure the availability of up-to-date data for the decision-makers. This eventually leads to early detection of and timely action against outbreaks.[10]

Respondents in our study found the lack of adequate human resource for surveillance at the district level as concerning. Literature shows that for surveillance to be effective, adequate human resource is essential to undertake field-based surveillance and for efficient data reporting.[21] The resource deficiencies including

field-based staff have been shown to undermine effective disease surveillance.[19 22] Moreover, putting the burden of surveillance activities on healthcare staff engaged in service delivery negatively affects their motivation, performance, and consequently, the quality of reporting.[10 20 23]

Despite several advantages of contracting out health services that the literature notes, the structural challenges largely remain unaddressed.[24] One such challenge is poor coordination between stakeholders. Many respondents pointed out that having multiple stakeholders in the same district had been detrimental to disease surveillance due to a lack of coordination and ambiguous reporting lines. The flow of data is independent of the level of health facility defeating the notion of integrated disease surveillance. Poor coordination between stakeholders[21 22] and between different levels of health facilities[10] have been reported to adversely affect the data reporting for disease surveillance.

Many study respondents pointed out the need to strengthen active surveillance in the district as in the present system, the surveillance was largely hospital based where the data were being gathered from patients presenting to health facilities. Literature shows that the majority of people may opt not to show up for health seeking unless they develop serious symptoms.[20] This, in the event of a communicable disease, not only has the potential for the infection to spread but also leads to a delay in detecting an outbreak. Relying on passive surveillance only is often dependent on factors including patients' awareness, health-seeking behaviour and socio-economic status and hence needs to be supplemented by some form of active surveillance.[23]

Paper-based reporting in our study came out as an important barrier to timely reporting. The need for transitioning from slow, staff-reliant and paper-based reporting to the digital mode of reporting is increasingly recognised in literature.[10 21] A study from India has demonstrated improved disease notification and enhanced data reporting due to transitioning to digital media.[25]

Except when a suspected or confirmed case of a disease is detected, data sent from health facilities are rarely reviewed at higher levels. Respondents in our study pointed out that there is more emphasis on data collection than its analysis and use. It is evident from the literature that providing regular feedback to facility staff on the data has been shown to act as a motivating factor and a performance boost.[10 22]

Study respondents emphasised the need for having local laboratory capacity to ensure the timely detection of disease outbreaks. Studies in literature have demonstrated that in absence of a local laboratory, rapid diagnostics kits may facilitate confirming outbreak until laboratory test results become available[23] thus preventing delays that may incur in the transfer of biological samples.[22]

The extent to which a health system can detect disease outbreaks is dependent on its capacity to capture patients' data. In Pakistan, the private health sector caters to most of the population's healthcare needs. However, it is largely unregulated in Sindh province and in Pakistan in general, leading to a lack of integration of its patient data with the public sector. Studies show that the poor private sector engagement in disease surveillance is an important issue hampering the surveillance efforts in many countries including India,[22 25] Iran[26] and China.[23]

## LIMITATIONS OF THE STUDY

Our study was conducted in a predominantly rural district of Pakistan that is already challenged in terms of resource availability. Hence, when generalising the study findings, these may be interpreted in a similar context. Due to the researchers' experience and familiarity with the health system of the study district, the possibility of contamination of study results with researchers' own perceptions cannot be completely eliminated, however, objectivity was ensured by note-taking during interviews and the use of interviews' audio recordings to ensure accuracy in data reporting.

## RECOMMENDATIONS

The lack of directions from the provincial level regarding infectious disease surveillance necessitates the need for formulation of policy guidelines outlining not just the technical aspects of surveillance but also ensuring adequate resource planning and allocation to establish and sustain effective infectious disease surveillance at the district level. To ensure disease surveillance in the district, there is a need to (1) address resource requirements including adequate budget and human resource; (2) engage with the private health sector to capture maximum data of patients presenting to health facilities; (3) build active surveillance into the existing system by having designated human resource and (4) take advantage of the paperless system for data reporting to eliminate reporting delays and make real-time reporting system where data is instantly available after collection.

## CONCLUSION

We concluded that poor governance was perceived to lead to underutilisation of existing resources for surveillance whereas lack of a policy framework on surveillance was considered to lead to a poor investment of resources in surveillance infrastructure. The absence of resources and inadequate human resource was identified by respondents as the key determinant of delayed and inadequate reporting, leading to delayed detection of disease outbreaks. This was further aggravated by the absence of local laboratory capacity. The existing surveillance system was perceived as largely paper-based, slow and composed of hospital-based passive surveillance. The lack of private sector engagement in infectious disease surveillance was perceived as a significant gap.

**Acknowledgements**  We acknowledge all our participants for their cooperation during the conduct of this study.

**Contributors**  IN, SS and RAS contributed to the plan and design of the study. IN developed the interview guide which RAS and RH reviewed and provided feedback. IN led data collection and performed data analysis. IN, SS, RAS and RH participated in the interpretation of the results. IN drafted the manuscript. SS, RAS and RH contributed to revisions of the manuscript for intellectual content. All authors approved the final version of the manuscript. The corresponding author (IN) accepts full responsibility for the finished work and/or the conduct of the study, had access to the data, and controlled the decision to publish.

**Funding**  This work was supported by a grant from the WHO grant number 1070479-0.

**Competing interests**  None declared.

**Patient and public involvement**  Patients and/or the public were not involved in the design, or conduct, or reporting, or dissemination plans of this research.

**Patient consent for publication**  Not applicable.

**Ethics approval**  This study involves human participants and was approved by Ethics Review Committee Aga Khan University Karachi Pakistan, reference # ERC # 2020-5777-15184. Participants gave informed consent to participate in the study before taking part.

**Provenance and peer review**  Not commissioned; externally peer reviewed.

**Data availability statement**  All data relevant to the study are included in the article or uploaded as online supplemental information. All the data collected as part of this research study are reported in the manuscript.

**Author note**  Checklist for the appropriate reporting statement: we used COnsolidated criteria for REporting Qualitative research (COREQ) checklist while writing the manuscript to ensure rigour in reporting the data for qualitative research

**ORCID iDs**
Imran Naeem http://orcid.org/0000-0002-9487-1303
Sameen Siddiqi http://orcid.org/0000-0001-8289-0964

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
