## [Reviewer comments · BMJ Open]

ARTICLE DETAILS

TITLE (PROVISIONAL)	Exploring stakeholders' experiences and perceptions regarding barriers to effective surveillance of communicable diseases in a rural district of Pakistan – a qualitative study
AUTHORS	Naeem, Imran; Siddiqi, Sameen; Siddiqi, Amna Rehana; Hasan, Rumina

VERSION 1 – REVIEW

REVIEWER	Alma Adler Brigham and Women's Hospital, Division of Global Health Equity
REVIEW RETURNED	24-Aug-2022

GENERAL COMMENTS	This is a timely and interesting paper. Some suggestions for restructuring and presentation of results below. Background Minor points: Line 64. Do you mean toppled here? Line 66. Do you mean inept? or perhaps inadequate, ill-prepared, or ill-equipped? Methods: you say that you conducted in depth interviews, but then say that they lasted roughly 30 minutes. You may want to consider changing saying you conducted semi-structured interviews You interviewed 12 males and 3 females. Out of interest is this roughly the proportion of healthcare managers and providers? or did you select participants to make sure you were able to get the perspectives of females as well as males? Were the females both managers and providers? Is it possible to get some information about how long the providers/managers had been working in this health system? someone who has one year of experience with the health system may have different insights than someone with 20 years. The data collections section could benefit from being tightened up a bit (ie condensed). There is a fair bit of repetition here. Results were there any differences in responses based on informants jobs, backgrounds, etc? Throughout results you make statements where it is unclear if you are reporting what the informants told you, or if it is something you know or have read about. For example:
--

	"A digital system named Provincial Disease Surveillance and Response Unit (PDSRU) was developed a few years ago with the support of donor money. The PDSRU had all the communicable diseases of importance listed and the system was expected to be linked up with secondary hospitals in the province for regular data collection" Throughout your results, please make it clear if this came out of informant interviews, or if this was from documents (if so please cite) or elsewhere. Many sections could be expanded a little. For example in the CDC section you state it lacks any infrastructure etc. So what is there? Specific points: line 183: "developed a few years ago" is it possible to be less vague? Your point here would be different if it had been developed 2 years ago versus 10 years ago. Discussion: The discussion could use a bit more structure. I would concentrate the summary of findings in the first section, and then move more to comparisons with other studies. right now it goes back and forth . Some suggestions: I would expand the first paragraph to discuss your major findings. For example: -you may also want to include in the summary how respondents found reporting to be passive and delayed (not suited for outbreak detection) -the line at 349: "Having the district surveillance coordinator as the sole person responsible for surveillance activities in the district without any additional resources was considered the major weakness in the surveillance system by study respondents" could go in the first paragraph as a major finding of the study. limitations. I find the last sentence of the limitations a bit out of nowhere. On what basis are you making this statement? you may want to comment on reflexivity of the authors here You also may want to consider as your positions may have influenced informants responses here. Please include a final paragraph after the limitations with some recommendations. How can you use the information from your study?
--	---

REVIEWER	Manmeet Kaur PGIMER, School of Public Health
REVIEW RETURNED	22-Sep-2022

GENERAL COMMENTS	Dear Authors A very good attempt on using qualitative methods of data collection and analyses. Being reviewer my effort is to improve the manuscript especially the idea/research question and the methods which are core to any reasearch. Most of all, think for a minute that why did not
--

	use the quantitative methods? Use of methods clarify your approach and standpoint which needs to be placed upfront. What was the purpose of your study is another important aspect of qualitative research which is not emerging clearly. 1. The title: Its too long and having many extra words. In the manuscript you have mentioned that it is the perceptions and the barriers that you are trying to explore, but the title is perception about barriers which is confusing and needs to be changed. Consider my comments mentioned above while revising the title. 2. Abstract: The objective can be translated into purpose of the study. Why did you do this study and if journal allows mention the research question. Objective and intro are almost the same and shadowing the purpose. 3. Introduction: Not asking for much but follow point no. 2 to mention the knowledge gaps, why are you doing this study and using qualitative methods. 4. Methods: Some serious issues. The design is mentioned differently in Abstract and in manuscript. You are doing descriptive exploratory study, why not grounded theory as you are doing inductive thematic analyses? It is confusing for the readers, especially those who are not much familiar with qualitative methods. 5. Findings/ Results- Please read analyses and come up with interpretation. You have put themes as the phrases which is ok but inductive analyses helps in interpretation and makes the perception clearer. Is their any link between perceptions and barriers? Its still not clear why policy could not be implemented. I think policy review is the limitation of the study. Will be good if you give a table of codes, themes and quotes. 6. Discussion: will follow from the revised manuscript, needs more explanation on how this study is useful/ recomendations or what practical solutions it suggests as the study is suggesting some. The limitations of the study require clarity. What is it that you think could have been done better? Why could you not do that? Best wishes
--	--

VERSION 1 – AUTHOR RESPONSE

Reviewer: 1

Dr. Alma Adler, Brigham and Women's Hospital, Harvard Medical School

Comments to the Author:

This is a timely and interesting paper. Some suggestions for restructuring and presentation of results below.

Background

Minor points:

- Line 64. Do you mean toppled here?

Response: to bring clarity, the sentence is revised as: “The Health System of the country is overwhelmed by issues of...” line 66

- Line 66. Do you mean inept? or perhaps inadequate, ill-prepared, or ill-equipped?

Response: to bring clarity, the sentence is revised as: “...resulting in a disease surveillance system that is ill-equipped at ...” lines 67-68

Methods:

- you say that you conducted in depth interviews, but then say that they lasted roughly 30 minutes. You may want to consider changing saying you conducted semi-structured interviews

Response: In the light of the reviewer's comment regarding interview duration, a discussion with field team members/note-takers present during interviews was done, and the duration is revised and agreed upon as 40 – 50 min per interview (line 143). This was an oversight as the actual interviews were longer than what was reported in the earlier draft of the manuscript.

For in-depth interviews, the interview guide used contained open-ended questions with probes, hence there was no structure/options for any of the questions.

- You interviewed 12 males and 3 females. Out of interest is this roughly the proportion of healthcare managers and providers? or did you select participants to make sure you were able to get the perspectives of females as well as males? Were the females both managers and providers?

Response: Yes, that is roughly the proportion of male versus female healthcare providers (doctors). Being a rural district, it is challenging for female doctors to be stationed at remotely located healthcare facilities. They will have to travel farther distances, and road infrastructure is poor/non-existent in these areas. It also brings concerns related to their safety. Hence most female doctors are stationed at the only secondary (district headquarter) hospital which is a relatively urban section of the study district. The majority of the healthcare facilities in the district are in rural areas with male doctors. Hence, we included female doctors also to explore their experiences and perceptions.

A healthcare manager's job involves travelling to remote rural facilities and rural communities hence all healthcare managers in the district are males.

Is it possible to get some information about how long the providers/managers had been working in this health system? someone who has one year of experience with the health system may have different insights than someone with 20 years.

Response: information regarding the experience of study participants has been added, lines 123-124.

The data collections section could benefit from being tightened up a bit (ie condensed). There is a fair bit of repetition here.

Response: as suggested, the section has been condensed and repetition has been removed, lines: 129-143.

Results

were there any differences in responses based on informants jobs, backgrounds, etc?

Response: The study respondents were native/belonged to the study district and had an experience of 5 – 30 years. Based on having the same background and experience with the health system of the same district, the authors didn't find differences in their responses.

Throughout results you make statements where it is unclear if you are reporting what the informants told you, or if it is something you know or have read about. For example:

"A digital system named Provincial Disease Surveillance and Response Unit (PDSRU) was developed a few years ago with the support of donor money. The PDSRU had all the communicable diseases of importance listed and the system was expected to be linked up with secondary hospitals in the province for regular data collection"

Response: As suggested, throughout the results section revisions are done to ensure clarity. The mentioned statement has been revised for clarity, line: 197

Throughout your results, please make it clear if this came out of informant interviews, or if this was from documents (if so please cite) or elsewhere.

Response: As suggested, throughout the results section revisions are done to ensure clarity, lines: 206, 213, 222, 238, 247, 259, 269, 276, 283-284, 287, 292, 297, 309, 311, 320, 327, 334, 340.

Many sections could be expanded a little. For example in the CDC section you state it lacks any infrastructure etc. So what is there?

Response: as suggested details have been added regarding the point raised above, lines 227-230.

Specific points:

line 183: "developed a few years ago" is it possible to be less vague? Your point here would be different if it had been developed 2 years ago versus 10 years ago.

Response: As per suggestion, it is revised by mentioning the exact durations for PDSRU, DHIS, and CDC sections, lines: 198, 206, 223.

Discussion:

The discussion could use a bit more structure. I would concentrate the summary of findings in the first section, and then move more to comparisons with other studies. right now it goes back and forth.

Response: As per suggestion, the first paragraph of the discussion has been rewritten to include all the key findings, lines: 347-361.

Also, revisions to text in discussion section are done to bring clarity and make it succinct, lines: 362-364, 366-367, 377-378, 382-383, 385-386, 389, 390, 396-398, 399-499, 402, 404.

Some suggestions:

I would expand the first paragraph to discuss your major findings. For example:

-you may also want to include in the summary how respondents found reporting to be passive and delayed (not suited for outbreak detection)

Response: As stated in response to the preceding comment, the first paragraph of the discussion is revised to incorporate the finding mentioned in the comment.

-the line at 349: "Having the district surveillance coordinator as the sole person responsible for surveillance activities in the district without any additional resources was considered the major weakness in the surveillance system by study respondents" could go in the first paragraph as a major finding of the study.

Response: As per suggestion, the finding mentioned in the comment has been moved to the first paragraph of the discussion, lines: 351-353

limitations.

I find the last sentence of the limitations a bit out of nowhere. On what basis are you making this statement?

Response: In line with the suggestion, the last sentence has been omitted, limitations section has been revised slightly.

you may want to comment on reflexivity of the authors here

Response: as per suggestion, a few lines have been added, lines: 433-437

A section on researchers' reflexivity is revised in lines: 151-161, the sequence is as per the journal's requirement.

You also may want to consider as your positions may have influenced informants responses here.

Response: as per suggestion, A section on researchers' reflexivity is revised in lines: 151-161.

Please include a final paragraph after the limitations with some recommendations. How can you use the information from your study?

Response: As per suggestion, a recommendations section has been added after the conclusion section, lines: 450-460

Reviewer: 2
Dr. Manmeet Kaur, PGIMER

Comments to the Author:

Dear Authors

A very good attempt on using qualitative methods of data collection and analyses. Being reviewer my effort is to improve the manuscript especially the idea/research question and the methods which are core to any research. Most of all, think for a minute that why did not use the quantitative methods? Use of methods clarify your approach and standpoint which needs to be placed upfront. What was the purpose of your study is another important aspect of qualitative research which is not emerging clearly.

Response: Dear reviewer, your comments are highly appreciated as these helped us, as authors, in critically appraising and revising the current version of our manuscript.

As per suggestion, revision in the text is done to better reflect the study rationale and answer why the study is done as qualitative research. lines: 86 – 95.

1. The title: Its too long and having many extra words. In the manuscript you have mentioned that it is the perceptions and the barriers that you are trying to explore, but the title is perception about barriers which is confusing and needs to be changed. Consider my comments mentioned above while revising the title.

Response: The title has been revised as: “Exploring stakeholders’ experiences and perceptions regarding barriers to effective surveillance of communicable diseases in a rural district of Pakistan – a qualitative study”. Here ‘regarding’ has replaced ‘of’. The title and objectives have been aligned to keep it consistent. This is kept consistent in abstract lines: 20 – 22, and in the main text, lines: 98 – 100.

Regarding shortening the title, we looked at published research articles in the BMJ open and other journals of repute. A few examples from BMJ Open are presented below:

- i - Strategies and challenges in Kerala’s response to the initial phase of COVID-19 pandemic: a qualitative descriptive study
- ii - Black sickle cell patients’ lives matter: healthcare, long-term shielding and psychological distress during a racialised pandemic in England – a mixed-methods study
- iii - Cross-sectional questionnaire study of the experiences of community pharmacists in Northern Ireland during the early phases of the COVID-19 pandemic: preparation, experience and response

All these titles range between word count of 17 – 24 and mention the study objective, study site and study design. When formulating a good title for our manuscript, we did so in line with the journal’s standard. Our title is 22 words in length which, considering the examples above, is in the range of word count for title length. We have tried making it succinct and removing any redundancies.

2. Abstract: The objective can be translated into purpose of the study. Why did you do this study and if journal allows mention the research question. Objective and intro are almost the same and shadowing the purpose.

Response: As per the journal’s requirement, we have given an objective. However, we have added a description in lines 96 – 98 which talks about the purpose of doing this study.

3. Introduction: Not asking for much but follow point no. 2 to mention the knowledge gaps, why are you doing this study and using qualitative methods.

Response: As mentioned in response to point 2, revision has been made in lines: 86 – 95.

4. Methods: Some serious issues. The design is mentioned differently in Abstract and in manuscript. You are doing descriptive exploratory study, why not grounded theory as you are doing inductive thematic analyses? It is confusing for the readers, especially those who are not much familiar with qualitative methods.

Response:

-Regarding study design, as per suggestion, “qualitative descriptive exploratory design” is uniformly mentioned in abstract line: 23 and in the main manuscript’s Methods section line: 103.

-Regarding the use of grounded theory

The grounded theory implies a data collection process that is iterative, whereby the researcher collects data, analyses it, and goes back to collect further data. These iterations keep going on until the researcher generates a (new) theory.

Descriptive exploratory design in qualitative research does not intend to generate a new theory and hence is non-iterative. Since researchers of this study didn’t intend to generate a new theory, hence they didn’t adopt an iterative approach for data collection. They used descriptive exploratory design hence it is mentioned in the manuscript.

-Regarding the use of inductive and deductive analysis

The analysis section of the manuscript has been expanded to include both deductive and inductive analysis

Researchers did deductive analysis because researchers’ experience and knowledge of the subject (health system and infectious diseases research) guided both the process of data collection and data analysis.

Researchers used inductive analysis because during data collection they explored new knowledge, so they used inductive analysis to inform the process of theme creation.

The revisions are mentioned on lines 165 – 170.

5. Findings/ Results- Please read analyses and come up with interpretation. You have put themes as the phrases which is ok but inductive analyses helps in interpretation and makes the perception clearer. Is there any link between perceptions and barriers? It’s still not clear why policy could not be implemented. I think policy review is the limitation of the study.

Will be good if you give a table of codes, themes and quotes.

Response:

-Regarding the link between perceptions and barriers: as clarified in response to point# 1, the authors are interested in exploring experiences and perceptions regarding barriers to disease surveillance.

Results show themes that represent various factors that in the experiences and perceptions of study respondents act as barriers to effective disease surveillance.

Some

- As per the reviewer’s comment, where appropriate the results have been revised/highlighted (yellow) or the line numbers are mentioned below to point out interpretations. Lines: 187-192, 213-217, 227-230, 238-242, 251-254, 263-268, 271-273, 279-285, 292-294, 300-305, 311-316, 335-339.

Further, interpretations of the results are done in the discussion section. The discussion section has been revised to include key findings in the first paragraph and a comparison of studies from the literature is done to reflect on similarities and differences.

-Regarding why policy could not be implemented, text on lines 362 – 367 has been revised to address and respond to the reviewer’s comment.

-Regarding policy review, as per study respondents, policy guidelines regarding disease surveillance were lacking at the provincial level. This is a finding of our study. The authors did not intend to do a review of policy documents

-As per suggestion, table 1 showing themes and codes has been added.

6. Discussion: will follow from the revised manuscript, needs more explanation on how this study is useful/ recommendations or what practical solutions it suggests as the study is suggesting some. The limitations of the study require clarity. What is it that you think could have been done better? Why could you not do that?

Best wishes

Response:

-As per suggestion, a recommendation section is added, lines: 450-460.

-Regarding study limitations, keeping in view that this is a qualitative study design, it is prone to researchers' bias. How researchers tried to minimize such bias, an explanation has been added in section "researchers' reflexivity", lines: 150-160

VERSION 2 – REVIEW

REVIEWER	Manmeet Kaur PGIMER, School of Public Health
REVIEW RETURNED	09-Oct-2022
GENERAL COMMENTS	A good revision.